# A Two-Stage Optimal Dispatching Model for Micro Energy Grid Considering the Dual Goals of Economy and Environmental Protection under CVaR

**Jun Dong, Yaoyu Zhang \*, Yuanyuan Wang and Yao Liu**

Department of Economic Management, North China Electric Power University, Beijing 102206, China; dongjun@ncepu.edu.cn (J.D.); 120192206002@ncepu.edu.cn (Y.W.); 120192206101@ncepu.edu.cn (Y.L.)
\* Correspondence: 120192206012@ncepu.edu.cn

**Abstract:** With the development of distributed renewable energy, a micro-energy grid (MEG) is an important way to solve the problem of energy supply in the future. A two-stage optimal scheduling model considering economy and environmental protection is proposed to solve the problem of optimal scheduling of micro-energy grid with high proportion of renewable energy system (RES) and multiple energy storage systems (ESS), in which the risk is measured by conditional value-at-risk (CVaR). The results show that (a) this model can realize the optimal power of various energy equipment, promote the consumption of renewable energy, and the optimal operating cost of the system is 34873 USD. (b) The dispatch of generating units is different under different risk coefficients $\lambda$, which leads to different dispatch cost and risk cost, and the two costs cannot be optimal at the same time. The risk coefficient $\lambda$ shall be determined according to the degree of risk preference of the decision-maker. (c) The proposed optimal model could balance economic objectives and environmental objectives, and rationally control its pollutant emission level while pursuing the minimum operation costs. Therefore, the proposed model can not only reduce the operation cost based on the consideration of system carbon emissions but also provide decision-makers with decision-making support by measuring the risk.

**Keywords:** micro-energy grid; renewable energy source; energy storage systems; economic and environmental protection; multi-objective decision; conditional value-at-risk

## 1. Introduction

### 1.1. Background and Motivation

With the development of national economies and the acceleration of urbanization, related problems in the field of energy have become increasingly prominent. Unreasonable use of fossil energy increases the emission of harmful substances such as $CO_2$ and nitrogen oxides. Therefore, it has become a general consensus of all countries in the world to seek clean energy to replace fossil energy and build a clean, low-carbon, and efficient energy system [1]. In recent years, under the guidance of the policy, China has vigorously developed the wind and solar renewable energy industry to solve the energy problem, but the large-scale development of renewable energy has caused a large number of wind and photovoltaic power curtailment problems [2]. With the dual challenges of global energy crisis and environmental pollution, the existing energy production and consumption patterns can hardly meet the needs of social development.

In 2016, The National Development and Reform Commission of China put forward "Guidelines on Promoting the Development of Internet plus Smart Energy", which pointed out that the construction of multi-energy coordinated energy grid should be strengthened, and the coupling interaction and comprehensive utilization of different types of energy such as electricity, gas, heat and cold should be carried out [3]. MEG is a kind of intelligent energy comprehensive utilization area grid, which has a higher proportion of renewable

energy access. It can also achieve basic balance between local energy production and energy use load through energy storage and optimal configuration, realize the multi-complementary of various distributed energy sources such as wind, photovoltaic, and natural gas, and flexibly interact with the public power grid according to needs [4]. It can absorb wind, photovoltaic, natural gas and other distributed energy through energy storage, conversion and optimal configuration, and meet energy load demands of electricity, heat, cold, and gas in coordination. It is of great significance to improve energy efficiency, develop renewable energy and reduce air pollution. The micro-energy grid will become an important new model of distributed energy utilization in the future [5]. With the gradual opening of the electricity market, the participation of the micro-energy grid in the spot market of electricity will be necessary. Therefore, the construction of the decision-making optimization model of the micro-energy grid in the spot market of electricity will become the key problem faced by the energy system.

### 1.2. Literature Review

MEG, as an effective form of distributed generation system integration, solves the problem of large-scale integrated application of renewable energy [6], and it is an important part of the development of energy internet terminal power supply [7]. Nowadays, many studies focus on the construction of optimal scheduling model of MEG to realize the optimal economic operation. However, since China has recently pledged to achieve carbon emission peak and neutrality by 2030 and 2060 respectively [8], it is required that the optimization of economic and environment dual objectives should be comprehensively considered in the optimization and scheduling of MEG under this background. Besides, when dealing with renewable energy generation, demand uncertainty and system operation risks are some of the important challenges in micro-energy grid dispatching [9]. The following is an overview of some of the methods used in previous studies to solve related problems.

At present, there have been many studies on the optimization of MEG scheduling and economic operation, most of which take the optimization of economic operation of MEG as the goal to build the optimal scheduling model [10–13]. However, the operation of MEG is complex and changeable, and it is difficult for the traditional single objective optimization to meet the needs of various aspects. Therefore, many studies have focused on multi-objective optimization. Reference [14] took minimizing the operation cost, carbon emission cost and primary energy conversion cost as the optimization goal, and adopts multi-objective optimization to coordinate different aspects of the operation of micro-energy network. Reference [15] established a multi-objective stochastic programming model of a micro-energy grid with minimum life-cycle cost and minimum carbon emission. Reference [16] set up a multi-objective optimization scheduling model of micro-energy grid based on typical scenarios based on the comprehensive consideration of economy, environment, and energy. Reference [17] studied the multi-objective optimal configuration of the key equipment capacity of a micro-energy grid with the goal of minimizing the whole life cycle cost and annual $CO_2$ emission.

Due to the uncertainty of wind power and photovoltaic, in order to ensure the stability and comprehensive benefit of multi-objective optimization, most studies have adopted robust optimization and stochastic optimization to deal with the uncertainty. Most researchers applied robust optimization to the treatment of uncertainty and related parameters of renewable energy [18], which has proven to be an effective method to ensure the reliability of micro-energy grid [19–22]. Among them, reference [19] proposed a robust optimization model of micro-energy network considering uncertainty, and sought for balance between the economy and robustness of micro-energy grid operation. Reference [20] proposed a robust optimization model based on the uncertainty of wind power and multi-demand response program based on the day-ahead dispatching stage and real-time adjustment stage. Reference [21] proposed a collaborative operation method of residential multi-micro-energy grid based on two-stage adaptive robust optimization, and deduced a scheduling scheme to minimize the operating cost under the uncertain realization of photovoltaic

output. In order to solve the uncertainty caused by intermittent renewable energy and random load, reference [22] proposed a comprehensive scheduling method based on robust multi-objective optimization.

However, robust optimization has some disadvantages, that is, the scheduling decision is too conservative, and comes at the cost of economic benefits [23]. Therefore, a stochastic optimization method has been proposed to solve this problem. Reference [24] applied stochastic programming to the optimal planning of the location and size of distributed generators in the island public microgrid under large-scale grid interference. Reference [25] proposed a hybrid microgrid optimization model based on mixed integer linear programming, and solved the problem based on stochastic optimization method. Reference [26] combined the chance-constrained stochastic optimization with big data analysis and applied them to the micro-grid energy management system. Reference [27] used stochastic optimization framework to solve the energy scheduling problem of micro-energy grid with stochastic renewable energy generation and vehicle activity mode. A stochastic multi-objective model for optimal energy exchange optimization of network microgrid is constructed, considering the uncertainty of load consumption and renewable energy generation [28].

In the process of optimizing the scheduling of micro-energy grid, the risks in the system also need to be considered. In recent years, some literature has proposed risk-based management methods, such as value-at-risk (VaR). However, VaR is an incoherent risk measure, lacking convexity and coherence, which makes it unpopular in practice [29]. In contrast, conditional value-at-risk (CVaR) is a coherent measure of risk that quantifies risks beyond VaR. It makes many large-scale calculations practical through linear programming techniques. Moreover, it has been applied to a number of grid-related problems to obtain optimal energy control recently [30]. Micro energy network operators apply CVaR in risk aversion decision to explain the uncertainty of power generation and electricity price of intermittent photovoltaic power generation system and to measure the risks caused by wind and photovoltaic abandonment, loss of load, and failure to provide energy or auxiliary services [31–34].

According to the summary of the above literature, it is obvious that the previous studies still have some deficiencies, mainly including the following three points. First of all, in terms of research objectives, most of them take the optimal economic operation of micro-energy grid as the goal to build the optimal scheduling model, without considering the dual objectives of economy and environment. Secondly, in terms of research methods, most studies adopt the idea of robust optimization, and the results are relatively conservative. Finally, in terms of risk consideration, some studies lack relatively reasonable and effective risk estimation.

### *1.3. Contributions and Organization*

In order to make up for the defects of existing research, this article put forward a two-stage optimal scheduling model for micro-energy grid with CVaR and economic and environmental binocular targets. It is based on the basic framework of MEG, and takes into account the dual objectives of economy and environment as well as the risks that the system may face during operation. The main innovations of this paper are as follows:

1.  A two-stage optimal dispatching model is proposed. In the first stage (hourly time scale), the forecast value of renewable energy output is input, and the daily dispatch plan of each equipment is formulated to minimize the pre-dispatch cost. In the second stage (15-min time scale), the components in the MEG are optimized and adjusted to minimize the unbalanced cost between the day-ahead and real-time stages, and stochastic optimization is used to deal with the uncertainty caused by renewable energy.

2. Both economic and environmental goals are considered. The optimal scheduling model under the comprehensive consideration of multiple objectives is constructed. In terms of the solution method, this paper converts the environmental protection objective into the economic objective by means of a unified dimension.

3. A two-stage optimal scheduling model for MEG based on CVAR, which can not only measure the system operation risk caused by the fluctuation of renewable energy, but also make a tradeoff between risk and cost by adjusting the risk preference coefficient. With proposed model and parameters on confidence level and risk preference, the system operators can choose the operation strategies properly.

4. A complex MEG with multi-energy supply and multi-energy consumption is constructed, which is equipped with the ESS including battery energy storage, heat storage tank, and ice energy storage, covering most of the system's energy supply and demand characteristics. The energy optimal dispatching model for the system has an unexceptionable universality, and can be applied to other types of systems.

The rest of the article is arranged as follows. The second section describes the main problems studied in this paper, the third section constructs the mathematical model of the micro-energy network, and the fourth section constructs and solves the mathematical model of the operation of the micro-energy grid. In Section 5, the rationality of the established model is verified by example analysis, and the final conclusion is given in Section 6. Finally, the abbreviations and acronyms are in Abbreviation.

## 2. Problem Description

This section briefly describes the problems, including the components of networked MEG, the proposed operational strategy and the basic assumptions in this paper.

### 2.1. Components of Networked MEG

The basic components of the MEG consist of RES (i.e., photovoltaics system and wind turbine), gas turbines, gas boilers, absorber chiller, power loads (i.e., both controllable and non-controllable loads), and storage batteries, heat storage tanks, ice storage chillers: three types of energy storage systems, hereinafter referred to as the ESS. RES can provide an extremely low marginal cost of electricity supply, with little or no greenhouse gas production. Gas turbines and gas boilers can provide a stable energy supply to meet the energy demand when RES is scarce. ESS can be adjusted by charging/discharging strategy to achieve the balance of energy in the time sequence, to alleviate the peak and valley difference of energy demand. Interruptible loads can maintain the energy supply and demand balance in extreme cases by reducing the energy demand on the load side.

In terms of gas consumption, there is no gas production in the MEG, all the gas needed is supplied by external gas companies, and there is only one-way purchase behavior between gas companies. Moreover, there is only one-way buying between MEG and gas companies. In terms of power interaction with the MGC, the MEG purchases power from large grids when the system power supply is insufficient and sells power to large grids when the system power supply has a surplus. Under market rules, electricity is bought at a higher price and sold at a lower price.

In China, thermal power is the main source of electricity, and $CO_2$, $SO_2$, and $NO_X$ will be produced by the electricity purchased from the main network. At the same time, a certain amount of $CO_2$ is produced in the operation of gas boilers and internal combustion engines in the system. The cost of treating these contaminated gases should be taken into account. From the point of view of MEG, the general goal of grid-connected mode is to minimize the operating cost or maximize the total benefit under certain operating constraints.

*2.2. Proposed Strategy and Assumption*

In this paper, a two-stage energy optimal scheduling model is proposed for a networked MEG with high renewable penetration. The model also considers the economic and environmental objectives with different dimensions, and solves the problem by converting the environmental objectives into economic objectives (system integrated dispatching costs) by unifying the dimensions [35]. The overall objective is to minimize the dispatching cost of networked MEG in grid connected modes and predefine the revenue risk into a certain level.

At the first stage, a day-ahead hourly scheduling is formulated for networked MEG. In this stage, the optimization objective is to minimize its operation cost. The problem is formulated as a deterministic issue without considering the uncertainty in the MEG, so the decision variables and constraints are not related to the scene. According to the forecasted RES output power, electrical load and electricity price determine the commitment status of energy supply equipment, charging/discharging status of ESSs, and the exchanged power between the MEG and utility grid.

At the second stage, a real-time dispatch is executed to balance the dynamic random fluctuations of RES at 15 min temporal resolution. Choosing the minimum deviation cost caused by the output fluctuation of RES is the objective function of this stage. The real-time stage represents the real-time operation process under different RES power scenarios, and the decision variables and constraints in this stage are related to the scenarios.

The risk of MEG with a high proportion of RES comes from the fluctuation of RES output, which is embodied in waste wind/light cost, load reduction cost, and reserve capacity scheduling cost. If you want to achieve the least risk, you need to increase the unit reserve, that is, in the pre-adjustment stage, to leave a larger rotation reserve range. Although this can improve the reliability of the system and reduce the occurrence of wind rejection and load loss, it will lead to a low utilization rate of rotating reserve capacity and increase unnecessary economic losses. Therefore, it is necessary to comprehensively consider the economic benefits of the rotating reserve and the CVaR value that the system may suffer losses when optimizing the rotating reserve.

The mathematical modelling and detailed steps of the two-stage model are described in Section 4.

## 3. Mathematical Model of Micro-Energy Grid

The micro-energy grid includes four energy forms: cooling, heating, electricity, and gas, and has the characteristics of diverse load types and abundant energy supply equipment. The micro-energy grid studied in this paper mainly includes five links: energy input, energy conversion, energy collection and distribution, energy storage, and energy output. Among them, energy input includes utility grid, natural gas grid, and distributed power sources, and energy conversion. Collection and distribution links are gas-heating (cooling) coupling and electric-heating (cooling) coupling. Gas-heat (cold) coupling equipment includes gas boilers and gas turbines, based on which the energy is converted and utilized by using waste heat recovery devices, absorption chillers, heat exchangers, and other equipment. Electric-heating (cooling) coupling equipment including electric boilers and electric chillers, can realize heating (cooling) driven by electricity; energy storage includes electrical/heating/cooling energy storage devices; energy output includes electrical/heating/cooling load. The system structure and energy flow process are shown in Figure 1, which details the composition of the distributed micro-sources in the micro-energy grid and the energy coupling and transformation relationships between devices.

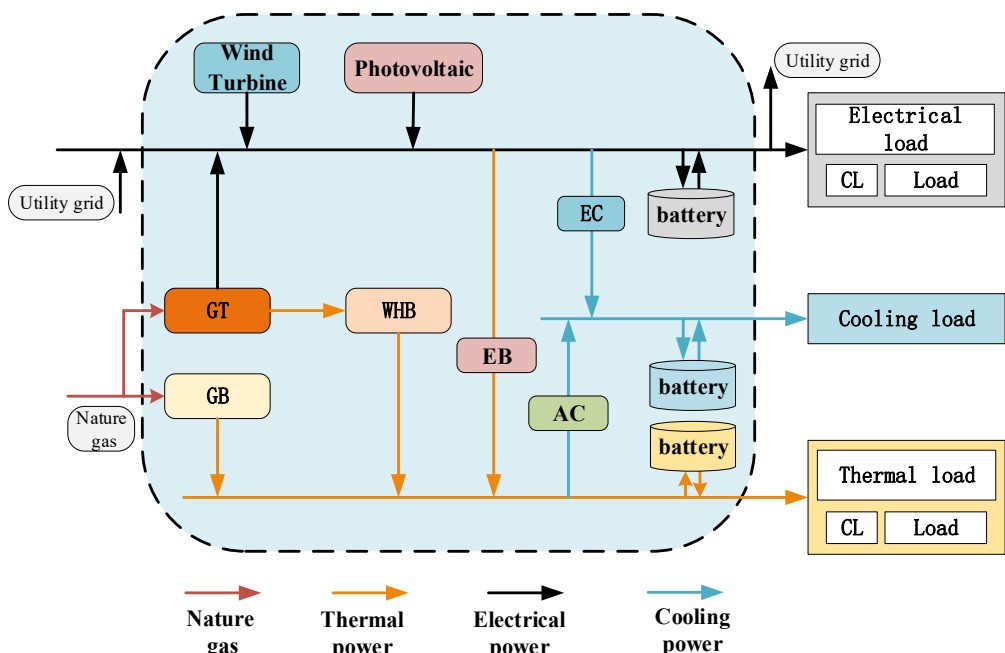

**Figure 1.** Micro energy grid system structure.

*3.1. Renewable Energy Power Generation*

1.    Wind Turbine

Wind energy is a kind of clean energy with huge reserves, which has been greatly valued by countries all over the world. Wind power is a way to convert wind energy into electrical energy. Because wind speed is affected by many factors such as temperature, air pressure, altitude, latitude, surface conditions, obstacles, etc., which have strong volatility and randomness, wind power output also presents uncertainty. Ref. [36] points out that the wind speed distribution obeys the Weibull distribution. This paper uses the method of moment estimation and time series to calculate the shape parameters and scale parameters of the Weibull distribution, and uses Monte Carlo simulation, combined with the Weibull randomly generated number in MATLAB, to calculate the corresponding wind output and generate wind power output scenarios. Linearized modeling of wind power and wind speed:

$$P_{wt} = \begin{cases} 0 & 0 \le v \le v_{ci} \\ \frac{v - v_{ci}}{v_r - v_{ci}} & v_{ci} \le v \le v_r \\ P_r & v_r \le v \le v_{co} \\ 0 & v \ge v_{co} \end{cases} \tag{1}$$

In Equation (1), $P_{wt}$ is the actual wind output power, kW. $Pr$ is the rated power of the wind turbine, kW. $v$ represents the actual wind speed of the wind turbine; m/s, $v_{ci}$, $v_r$, and $v_{co}$ respectively represent the cut-in wind speed, rated wind speed, and cut-out wind speed of the wind turbine, in m/s. The relationship between the output of the wind power and the wind speed is shown in Figure 2.

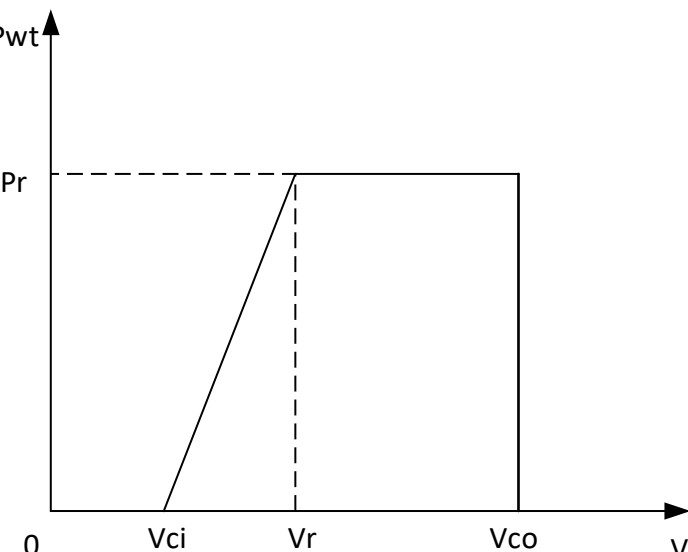

**Figure 2.** Wind turbine output power curve.

2. Photovoltaic

Solar energy is the most abundant and widely distributed renewable energy in the world. Photovoltaic power generation is a way to directly convert solar energy into electrical energy. It is a clean energy source just like wind power and has great development prospects. The photovoltaic power is related to solar irradiance and ambient temperature. In this paper, to simplify the calculation, only the influence of solar irradiance is considered. The solar irradiance has the same uncertainty as the wind speed, and generally obeys the normal distribution [37]. The average value and variance of the solar irradiance are obtained by moment estimation of historical data, and then the photovoltaic power is calculated in this paper. The mathematical model of photovoltaic power generation is usually expressed as:

$$P_{pv} = \zeta \cdot A_P \cdot \eta_P \tag{2}$$

In Equation (2), $P_{pv}$ represents the output power of the photovoltaic, kW; $\zeta$ represents the actual solar irradiance, kW/m$^2$; $A_P$ is the solar area of the photovoltaic panel, m$^2$; and $\eta_P$ is the photoelectric conversion efficiency of the photovoltaic panel.

*3.2. Energy Conversion Equipment*

1. Gas turbine

Due to their fast start-stop speed and lower operating cost, gas turbines can provide power for the MEG during periods of low RES output and ensure the stable operation of the system. The MEG with gas turbines uses the heating energy of natural gas combustion to drive the gas turbines to generate electricity, and the waste heat generated is recovered by the waste heat boiler to supply the heating load demand. The mathematical model is:

$$F_{gt,t} = \frac{P_{gt,t} \cdot \Delta t}{L_{hvng} \cdot \eta_{gt}} \tag{3}$$

$$H_{gt,t} = \frac{P_{gt,t} \cdot (1 - \eta_{gt} - \eta_L)}{\eta_{gt}} \tag{4}$$

$$H_{whb,t} = H_{gt,t} \cdot \eta_{whb} \tag{5}$$

where $F_{gt,t}$ is the amount of natural gas consumed by the gas turbine, and $P_{gt,t}$ is the electrical power output by the gas turbine, kW. $L_{hvng}$ is the low heating potential value of the natural gas. $\eta_{gt}$ is the power generation efficiency of the gas turbine, and $\Delta t$ represents an operating period. $H_{gt,t}$ is the exhaust heat of the gas turbine, kW, and $\eta_L$ is the heat

dissipation loss coefficient of the gas turbine. $H_{whb,t}$ is the heating energy provided by the waste heat of the gas turbine flue gas, kW; $\eta_{whb}$ is the heating coefficient of the waste heat recovery device.

2.  Gas boiler

Gas boiler is a device that consumes natural gas and converts it into heating energy, and realizes energy transfer by heating the water in the boiler [37]. Compared with coal-fired boilers, they produce less pollutants and have higher heating efficiency, which can be used as a supplementary heating source for the micro-energy grid. The mathematical model is as follows:

$$H_{b,t} = \eta_{gb} \cdot F_{b,t} \tag{6}$$

In Equation (6), $H_{b,t}$ is the heating power of the gas boiler, kW. $F_{b,t}$ is the heating value of the gas consumed in the period $t$, kW. $\eta_{gb}$ is the heat production efficiency of the gas boiler.

3.  Electric boiler

$$H_{eb,t} = \eta_{eb} \cdot P_{eb,t} \tag{7}$$

In Equation (7), $H_{eb,t}$ is the heating power of the electric boiler, kW. $P_{eb,t}$ is the electrical power consumed in the period $t$, kW. $\eta_{eb}$ is the heating-electric conversion efficiency of the electric boiler.

4.  Heat exchanger

The heat exchanger refers to the fact that heating energy can convert other heat energy into the heating value required by the user to meet the user's heating load [38], the mathematical model is as follows:

$$H_{he,t} = Q_{R,t} \cdot \eta_{he} \tag{8}$$

In Equation (8), $H_{he,t}$ is the heating power output by the heat exchanger, kW. $Q_{R,t}$ is the heating power entering the heat exchanger, kW. $\eta_{he}$ is the heating efficiency of the heat exchanger.

5.  Absorption chiller

With the development of combined cooling, heating, and electricity technology, absorption chillers have been widely used. The power source of this kind of chiller is heating energy. Its working principle is as follows: by using two solutions with different boiling points to form a set of heat transfer media, the heat transfer medium with lower boiling point is used as the refrigerant to achieve the evaporative cooling effect, and the heat transfer medium with higher boiling point is used as the refrigerant, which can absorb steam to realize the refrigeration cycle. The mathematical model can be expressed as:

$$Q_{ac,t} = H_{ac,t} \cdot \eta_{ac} \tag{9}$$

In Equation (9), $H_{ac,t}$ is the input heating power, kW. $Q_{ac,t}$ is the cooling power of the absorption chiller, kW., and $\eta_{ac}$ is the refrigeration coefficient of the absorption chiller.

6.  Electric chiller

An electric chiller is a device that converts electricity into cooling energy. It mainly uses electricity to realize energy transfer instead of directly producing cooling energy. It has a high energy efficiency ratio [39]. In the multi-energy system constructed in this paper, the electric chiller and the absorption chiller together form the cold energy source, which can not only improve the efficiency of the multi-energy system, but also can be used as a

peak-shaving equipment during the peak cooling load. Compared with absorption chillers, electric chillers have higher cooling efficiency. The mathematical model is expressed as:

$$Q_{ec,t} = P_{ec,t} \cdot \eta_{ec} \tag{10}$$

In Equation (10), $Q_{ec,t}$ is the cooling power, kW. $P_{ec}$ represents the electric power consumed by cooling, kW, and $\eta_{ec}$ is the cooling coefficient.

*3.3. Energy Storage Equipment*

1.  Battery energy storage

There is great uncertainty about the power of distributed power generation facilities (wind, photovoltaic) in the micro-energy grid, and the configuration of electricity storage equipment with appropriate capacity can better improve the quality and reliability of power supply [40]. This paper mainly considers the charging and discharging power and the current state of electricity storage. It does not make fine modeling of the internal circuits and components of electricity storage, and control strategy is not within the scope of this paper. The operating status of electricity storage is as follows:

$$W_{bt,t} = W_{bt,t-1}(1 - \sigma_{bt}) + \left(\eta_{bt}^{chr} \cdot P_t^{bt,chr} - P_t^{bt,dis}/\eta_{bt}^{dis}\right) \cdot \Delta t \tag{11}$$

In Equation (11), $W_{bt,t}$ is the amount of electricity stored by the battery at time $t$, $P_t^{bt,chr}$ and $P_t^{bt,dis}$ respectively represent the charging and discharging power of the electricity storage, $\sigma_{bt}$ is the energy self-loss rate of the battery, $\eta_{bt}^{chr}$ and $\eta_{bt}^{dis}$ represent the charging and discharging efficiency of the battery, respectively.

2.  Heating energy storage

Heating energy storage is mainly used to solve the problem of mismatch between heating load and heating energy supply. It can store excess heating energy during low heating load periods and release heating energy during peak heat load periods to achieve the effect of peak-shaving and valley-filling of heating load. For the coordinated management of electrical and heating loads, the basic model is consistent with that of electricity storage.

$$W_{tst,t} = W_{tst,t-1}(1 - \sigma_{tst}) + \left(\eta_{tst}^{chr} \cdot P_t^{tst,chr} - P_t^{tst,dis}/\eta_{tst}^{dis}\right) \cdot \Delta t \tag{12}$$

In Equation (12), $W_{tst,t}$ and $W_{tst,t-1}$ respectively represent the capacity of the heat energy storage device at $t$ and $t-1$. $P_t^{tst,chr}$ and $P_t^{tst,dis}$ respectively represent the heating storage and release power of the heating energy storage device. $\sigma_{tst}, \eta_{tst}^{chr}$, and $\eta_{tst}^{dis}$ represent heat loss rate and charge/discharge efficiency of heating energy storage, respectively.

3.  Cooling energy storage

Ice storage is used for refrigeration by electric chiller during the low period of electricity consumption at night, and the produced cold energy is stored in the form of ice. During the cooling load peak period, the ice is melted to release cooling energy and provide cooling to users to meet the cooling load demand [41]. The ice storage device is used as the cooling energy supply of the cooling storage device to assist the micro-energy grid, and its ice-making mode and ice melting mode are studied. The mathematical models of the ice storage device in the two operating modes are as follows:

The mathematical model in the ice-making operation mode is:

$$P_t^{it,ice} = \frac{C_{ic,t}}{a_1 C_{ic,t} + a_2} \tag{13}$$

In Equation (13), $C_{ic,t}$ is the ice making power of the ice storage device, $P_t^{it,ice}$ is the power consumption of the ice making of the ice storage tank in the ice storage device, $a_1$ and $a_2$ are the ice-making performance coefficients of the ice storage device.

The mathematical model in the ice melting operation mode is:

$$P_t^{it,melt} = \frac{C_{ice,t}}{W_{ice}} \cdot P_d \tag{14}$$

In Equation (14), $C_{ice,t}$ is the melting power of the ice storage device, $P_t^{it,melt}$ is the power consumption of ice melting in the ice storage tank of the ice storage device, $P_d$ is the rated ice melting power consumption of the ice storage device, and $W_{ice}$ is the rated ice melting refrigeration of the ice storage device power.

This paper mainly considers the ice-making and melting power of the ice-storage device and the current state of the cooling storage capacity, and describes the operating state of the ice-storage device with the value of the cooling storage state and the ice-making and melting power as variables [42].

$$W_{it,t} = W_{it,t-1}(1 - \sigma_{it}) + \left( \eta_{it}^{chr} \cdot P_t^{it,ice} - P_t^{it,melt} / \eta_{it}^{dis} \right) \cdot \Delta t \tag{15}$$

In Equation (15), $\sigma_{it}$ is the cold storage loss rate of the ice storage device, and $\eta_{it}^{chr}$ and $\eta_{it}^{dis}$ are the ice making and melting efficiency of the ice storage device, respectively.

## 4. Proposed Two-Stage Operation Model

Based on the basic model framework of the micro-energy grid proposed in the previous section, while considering system economy and environmental protection, and taking into account the risks that the system may face, a two-stage scheduling model including day-ahead and real-time is established. In the day-ahead stage, the system operator arranges the start-up and shutdown plan and energy dispatch plan of each unit based on the forecast of renewable energy power, with the goal of minimizing the pre-dispatch cost of the system. The real-time stage is mainly used to correct the forecast deviation, which represents the real-time operation process corresponding to different renewable energy power and load scenarios. The decision variables of the day-ahead stage are used as input variables in the real-time stage to participate in the optimization. In addition, CVaR is used to measure the risks faced in the operation process. The Figure 3 shows the overall framework of the two-stage optimization scheduling model constructed in this paper.

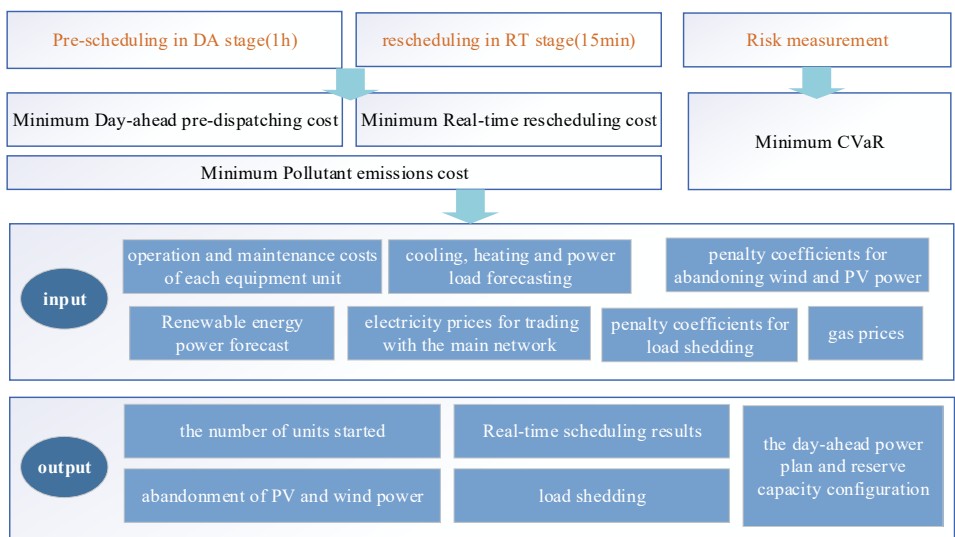

**Figure 3.** Two stage optimal scheduling model framework.

*4.1. Hourly Day-Ahead Optimal Scheduling Model*

4.1.1. Objective Function

The pre-dispatch model of the day-ahead stage aims at optimal system economy and guarantees certain environmental protection [43,44]. The day-ahead stage objective function is as follows:

$$\min f^{DA} = \sum_{t=1}^{T} \left[ \mu_1 \left( C_{ng,t} + C_{rm,t} + C_{spare,t} + C_{grid,t} \right) + \mu_2 C_{poll,K,t} \right] \cdot \Delta t \tag{16}$$

In Equation (16), $\mu_1$ and $\mu_2$ are the corresponding weights of system operating cost and environmental cost, respectively, and $\mu_1 + \mu_2 = 1$. The first half of the Equation (16) is the economic cost, $C_{ng,t}$, $C_{rm,t}$, $C_{spare,t}$, and $C_{grid,t}$ respectively represent gas cost, operation and maintenance cost, spare cost, and the cost of interacting with the utility grid, in USD. The second half of Equation (16) represents the emission cost, $C_{poll,K,t}$ is the emission cost of $K$ pollutants, USD; $K$ is the total number of pollutant emission types.

The calculation of $C_{ng,t}$ is given in Equation (17):

$$C_{ng,t} = \left[ \left( \sum_{\sigma=1}^{N_\sigma} F_{gt,t}^{\sigma} + \sum_{\tau=1}^{N_\tau} F_{b,t}^{\tau} \right) \cdot c_{ch4} \right] / L_{hvng} \tag{17}$$

In Equation (17), $F_{gt,t}^{\sigma}$ is the power of the gas turbine $\sigma$, kW; $F_{b,t}^{\tau}$ is the power of the gas boiler $\tau$, kW; $c_{ch4}$ is the unit cost of natural gas, USD/kWh.

The calculation of $C_{rm,t}$ is given in Equation (18):

$$
\begin{aligned}
C_{rm,t} = & \sum_{\sigma=1}^{N_\sigma} \left( P_{gt,t}^{\sigma} \cdot R_{gt,rm} + H_{whb,t}^{\sigma} \cdot R_{whb,rm} \right) + \sum_{\tau=1}^{N_\tau} H_{b,t}^{\tau} \cdot R_{b,rm} \\
& + H_{he,t} \cdot R_{he,rm} + H_{ac,t} \cdot R_{ac,rm} + P_{ec,t} \cdot R_{ec,rm} + \left( P_t^{bt,chr} + P_t^{bt,dis} \right) \cdot R_{bt,rm} \\
& + \left( P_t^{tst,chr} + P_t^{tst,dis} \right) \cdot R_{tst,rm} + \left( P_t^{it,chr} + P_t^{it,dis} \right) \cdot R_{it,rm}
\end{aligned} \tag{18}
$$

In Equation (18), $C_{rm,t}$ is the daily operation and maintenance cost of the internal energy equipment in the MEG, USD. $R_{i,rm}$ is the unit power operation and maintenance cost of the energy equipment, USD/kWh. $P_{gt,t}^{\sigma}$ is the electrical power of the gas turbine $\sigma$, kW; $H_{whb,t}^{\sigma}$ is the heating power of the waste heat boiler $\sigma$, kW. $H_{b,t}^{\tau}$ is the heat power of the gas boiler $\tau$, kW. $H_{he,t}$ is the heat exchanger power, kW. $H_{ac,t}$ is the absorption chiller power, kW. Finally, $P_{ec,t}$ is the power of the electric refrigerator, kW. $P_t^{bt,chr}$ and $P_t^{bt,dis}$ are the battery charge and discharge power, kW. $P_t^{tst,chr}$ and $P_t^{tst,dis}$ are the heating power of heat energy storage, kW. $P_t^{it,chr}$ and $P_t^{it,dis}$ are the cooling power of ice energy storage, kW.

The calculation of $C_{spare,t}$ is given in Equation (19):

$$C_{spare,t} = \sum_{\sigma=1}^{N_\sigma} \left( RU_{\sigma,t} \cdot \pi_{\sigma,t}^{RU} + RD_{\sigma,t} \cdot \pi_{\sigma,t}^{RD} \right) + \sum_{\tau=1}^{N_\tau} \left( RU_{\tau,t} \cdot \pi_{\tau,t}^{RU} + RD_{\tau,t} \cdot \pi_{\tau,t}^{RD} \right) \tag{19}$$

In Equation (19), $N_\sigma$ is the number of gas turbines participating in the standby plan, $RU_{\sigma,t}$ and $RD_{\sigma,t}$ are respectively the up and down standby scheduling capacity of the unit $\sigma$, kW; $\pi_{\sigma,t}^{RU}$ and $\pi_{\sigma,t}^{RD}$ are the upward and downward standby prices in the day-ahead stage, respectively, USD/kWh. $N_\tau$ is the number of gas boilers participating in the standby plan; $RU_{\tau,t}$ and $RD_{\tau,t}$ are respectively the up and down standby scheduling capacity of the unit $\tau$, kW, respectively; $\pi_{\tau,t}^{RU}$ and $\pi_{\tau,t}^{RD}$ are the up and down standby prices in the day-ahead stage respectively.

The calculation of $C_{grid,t}$ is given in Equation (20):

$$C_{grid,t} = \rho_{buy,t} \cdot P_{buy,t} - \rho_{sell,t} \cdot P_{sell,t} \tag{20}$$

In Equation (21), $\rho_{buy,t}$ and $\rho_{sell,t}$ are the unit electricity price for the micro-energy grid to purchase electricity from the utility grid and sell electricity to the utility grid during the dispatch period $t$, respectively, USD/kWh. $P_{buy,t}$ and $P_{sell,t}$ are the power that the MEG purchases from the utility grid and sells to the utility grid during the dispatch period $t$, kW.

The calculation of $C_{poll,K}$ is given in Equations (21) and (22):

$$C_{poll,K,t} = \sum_{k=1}^{K} (\gamma_k \cdot \xi_k \cdot POLL_{k,t}) \tag{21}$$

$$POLL_{k,t} = P_{buy,t} + \sum_{\sigma=1}^{N_\sigma} F_{gt,t}^\sigma + \sum_{\tau=1}^{N_\tau} F_{b,t}^\tau \tag{22}$$

In Equations (21) and (22), $\gamma_k$ is the cost of treating the $k$-th pollutant (USD/ton), and $\xi_k$ is the emission coefficient of the $k$-th pollutant (ton/MWh), $POLL_{k,t}$ is the output power of the pollutant generating equipment at time $t$.

4.1.2. Constraints

1. Equipment power constraints:

$$u_{i,t} P_i^{min} \leq P_{i,t}^{da} \leq u_{i,t} P_i^{max} \tag{23}$$

In Equation (23), $u_{i,t}$ is a binary variable representing the operating state of the gas turbine $i$. It is set to 1 when it is in the operating state, otherwise it is set to 0. $P_{i,t}^{da}$ is the actual power of equipment $i$, $P_i^{max}$ and $P_i^{min}$ are the maximum power and minimum power of equipment $i$, respectively.

2. Equipment start and stop constraints:

$$u_{i,t}^{off} = u_{i,t-1} - u_{i,t} + u_{i,t}^{on} \tag{24}$$

$$u_{i,t}^{off} + u_{i,t}^{on} \leq 1 \tag{25}$$

In Equations (24) and (25), $u_{i,t}^{on}$ and $u_{i,t}^{off}$ respectively represent the start and stop state variables of gas turbine and gas boiler at time $t$. When the unit $i$ starts at time $t$, $u_{i,t}^{on}$ takes 1, $u_{i,t}^{off}$ takes 0; when unit $i$ stops at time $t$, $u_{i,t}^{on}$ takes 0, and $u_{i,t}^{off}$ takes 1.

3. Climbing power constraint:

$$-RD_i \leq P_{i,t}^{da} - P_{i,t-1}^{da} \leq RU_i \tag{26}$$

In Equation (26), $RU_i$ and $RD_i$ are respectively the upward and downward climbing rate of equipment $i$.

4. Spinning reserve constraints:

$$R_{i,U,t} = min\left\{ RU_i, \left( P_i^{max} - P_{i,t}^{da} \right) \right\} \tag{27}$$

$$R_{i,D,t} = min\left\{ RD_i, \left( P_{i,t}^{da} - P_i^{min} \right) \right\} \tag{28}$$

In Equations (27) and (28), $R_{i,U,t}$ and $R_{i,D,t}$ are respectively the up and down standby scheduling capacity of the unit $i$ in time $t$.

5. Constraints of battery/heat/cold energy storage:

$$W_i^{min} \leq W_{i,t} \leq W_i^{max} \tag{29}$$

$$0 \leq P_t^{i,chr} \leq P_i^{i,max} * U_t^{i,chr} \tag{30}$$

$$0 \leq P_t^{i,dis} \leq P_i^{i,max} * U_t^{i,dis} \tag{31}$$

$$U_t^{i,chr} + U_t^{i,dis} \leq 1 \tag{32}$$

In Equation (29), $W_i^{min}$ and $W_i^{max}$ are the minimum and maximum energy storage capacity of energy storage equipment $i$. In Equations (30)–(32), $U_t^{i,chr}$ and $U_t^{i,dis}$ are the binary state variable representing the charging and discharging of the energy storage equipment $i$, and when charging $U_t^{i,chr}$ is set to 1, $U_t^{i,chr}$ is set to 0, and the opposite is used for discharge.

6.　Micro-energy grid and external grid interactive power constraints:

$$0 \leq P_t^{buy} \leq U_t^{buy} \cdot P^{buy,max} \tag{33}$$

$$0 \leq P_t^{sell} \leq U_t^{sell} \cdot P^{sell,max} \tag{34}$$

$$U_t^{buy} + U_t^{sell} \leq 1 \tag{35}$$

In Equations (33)–(35), $P_t^{sell}$ is the power sold by the MEG to the utility grid at time $t$, and $P_t^{buy}$ is the power purchased by the system from the utility grid at time $t$. Due to the limitation of the transmission grid capacity, $P^{buy,max}$ and $P^{sell,max}$ are the maximum values of the interaction power between the MEG and utility grid. $U_t^{buy}$ and $U_t^{sell}$ represent the binary variables of the power purchase and sale status of the system. $U_t^{buy}$ is set to 1, $U_t^{sell}$ is set to 0 when purchasing electricity, and the opposite is the case when selling electricity.

7.　Power balance constraints:

$$P_{pv,t} + P_{wt,t} + \sum_{\sigma=1}^{N_\sigma} P_{gt,t}^\sigma + P_t^{bt,dis} + P_t^{buy} - P_t^{sell} = P_{ec,t} + P_t^{bt,chr} + P_{eb,t} + P_{L,t} \tag{36}$$

$$\sum_{\tau=1}^{N_\tau} H_{b,t}^\tau + \sum_{\sigma=1}^{N_\sigma} H_{whb,t}^\sigma + H_{eb,t} + P_t^{bt,dis} = H_{ac,t} + P_t^{bt,chr} + H_{H,t}/\eta_{he} \tag{37}$$

$$COP_{ac} \cdot H_{ac,t} + COP_{ec} \cdot P_{ec,t} + P_t^{it,melt} = P_t^{it,ice} + Q_{C,t} \tag{38}$$

In Equations (36)–(38), $P_{L,t}$, $H_{H,t}$, and $Q_{C,t}$ are the electrical load, heating load, and cooling load at time $t$ respectively. $\eta_{he}$ is the heat exchange coefficient; $COP_{ac}$ and $COP_{ec}$ are the chiller coefficients of absorption chillers and electric chillers.

*4.2. 15-Minute Real-Time Dispatch Model*

4.2.1. Objective Function

In real-time dispatch, the dynamic fluctuations of RES are accommodated in the operation of MEG. The real-time rescheduling is an optimal rescheduling scheme with minimum rescheduling cost based on the pre-scheduling of MEG, which considers the deviation between forecasted and actual value of RES and different supply cost of backup energy. Note that the real-time dispatch interval could be any short uniform time interval. In this paper, the proposed dispatching interval is assumed to be 15 min, the time window of the dispatch covers 96 intervals (i.e., 24 h). Establish the objective function as shown in Equation (39):

$$\min f^{RT} = \sum_{t=1}^{ST} \left[ \mu_1 \left( C_{t,r} + C_{t,waste} + C_{t,cl} \right) + \mu_2 C_{poll,K,t} \right] \cdot \Delta t \tag{39}$$

In Equation (39), $ST$ is the total number of dispatch intervals in the real-time stage, $C_{t,r}$ is the standby cost of gas turbine and gas-fired boiler, $C_{t,waste}$ is the penalty cost of abandoning wind and PV power, and $C_{t,cl}$ is the penalty cost of electricity/heating/cooling load reduction. The unit of each variable in the real-time dispatch is consistent with the day-ahead dispatch, so it is not repeated.

The calculation of $C_{t,r}$ is given in Equation (40):

$$C_{t,r} = \sum_{\sigma=1}^{N_\sigma} \left( \pi_{\sigma,t}^{ru} \cdot r_{us,\sigma,t} + \pi_{\sigma,t}^{rd} \cdot r_{ds,\sigma,t} \right) + \sum_{\tau=1}^{N_\tau} \left( \pi_{\tau,t}^{ru} \cdot r_{us,\tau,t} + \pi_{\tau,t}^{rd} \cdot r_{ds,\tau,t} \right) \tag{40}$$

In Equation (40), $\pi_{\sigma,t}^{ru}$ and $\pi_{\sigma,t}^{rd}$ are the up and down standby prices adjusted by gas turbine for real-time electric power fluctuation; $r_{us,\sigma,t}$ and $r_{ds,\sigma,t}$ are gas turbine $\sigma$ up and down standby scheduling capacity at time $t$. $\pi_{\tau,t}^{ru}$ and $\pi_{\tau,t}^{rd}$ are the up and down standby prices adjusted by gas boiler for real-time heat power fluctuation; $r_{us,\tau,t}$ and $r_{ds,\tau,t}$ are gas boiler $\tau$ up and down standby scheduling capacity.

The calculation of $C_{t,waste}$ is given in Equation (41):

$$C_{t,waste} = K_P \cdot \left( P_{t,waste}^{pv} + P_{t,waste}^{wt} \right) \tag{41}$$

In Equation (41), $K_P$ represents the penalty cost of abandoning wind and PV power, $P_{t,waste}^{pv}$ and $P_{t,waste}^{wt}$ represent the amount of abandoned wind and PV power at time $t$.

The calculation of $C_{t,cl}$ is given in Equation (42):

$$C_{t,cl} = K_L \cdot P_{t,w}^{cl} + K_H \cdot H_{t,w}^{cl} + K_Q \cdot Q_{t,w}^{cl} \tag{42}$$

In Equation (42), $K_L$, $K_H$, and $K_Q$ represent the penalty cost of electrical, heating, and cooling load reduction respectively, $P_t^{cl}$, $H_t^{cl}$, and $Q_t^{cl}$ represents the reduction of electrical, heating, and cooling load at time $t$ respectively.

### 4.2.2. Constraints

1. Equipment rescheduling constraints:

$$P_{i,t}^{RT} = P_{i,t}^{da} + r_{i,us,t} - r_{i,ud,t} \tag{43}$$

$$0 \leq r_{i,us,t} \leq u_{i,t}^u R_{i,U,t} \tag{44}$$

$$0 \leq r_{i,ud,t} \leq u_{i,t}^d R_{i,D,t} \tag{45}$$

$$u_{i,t}^u + u_{i,t}^d \leq 1 \tag{46}$$

In Equations (43)–(46), the real-time power value of gas turbine and gas boiler is equal to the day-ahead power value plus (minus) the re-dispatching power value, and the rescheduling power cannot exceed the previous maximum standby capacity. $u_{i,t}^u$ and $u_{i,t}^d$ are binary variables indicating the equipment rescheduling status, that is, equipment $i$ can only be scheduled up or down at time $t$.

2. Climbing power constraint:

$$-R_{i,D,t} \leq P_{i,t+1}^{RT} - P_{i,t}^{RT} \leq R_{i,U,t} \tag{47}$$

In Equation (47), $R_{i,U,t}$ and $R_{i,D,t}$ are respectively the upward and downward climbing rate of equipment $i$.

3. Interruptible load constraints:

$$0 \leq P_t^{cl} \leq P_{\varphi,t,max}^{cl} \tag{48}$$

$$0 \leq H_t^{cl} \leq H_{\varphi,t,max}^{cl} \tag{49}$$

In Equations (48) and (49), $P_t^{cl}$ and $H_t^{cl}$ are the demand response quantities of electrical load and heating load at time $t$, respectively. In order to simplify the model, the interruptible load is treated as a continuous variable, while the discrete interruptible load is usually used in practice.

4.  Abandoning wind and PV constraints:

$$0 \leq P_{t,waste}^{pv} \leq \max\left\{0, P_t^{pv,RT} - P_t^{pv}\right\} \tag{50}$$

$$0 \leq P_{t,waste}^{wt} \leq \max\left\{0, P_t^{wt,RT} - P_t^{wt}\right\} \tag{51}$$

In Equations (50) and (51), $P_t^{pv,RT}$ and $P_t^{wt,RT}$ are the PV and wind power at time $t$ respectively in the real-time stage; $P_{t,waste}^{pv}$ and $P_{t,waste}^{wt}$ represent the amount of abandonment of wind and PV caused by the inability to absorb the rapid increase in wind and PV power in the real-time stage.

5.  Power balance constraint:

$$\sum_{\sigma=1}^{N_\sigma} \left( P_{gt,t}^{\sigma,RT} - P_{gt,t}^{\sigma} \right) = P_{t,waste}^{pv} + P_{t,waste}^{wt} - P_t^{cl} \tag{52}$$

$$\sum_{\sigma=1}^{N_\sigma} \left( H_{whb,t}^{\sigma,RT} - H_{whb,t}^{\sigma} \right) + \sum_{\tau=1}^{N_\tau} \left( H_{b,t}^{\tau,RT} - H_{b,t}^{\tau} \right) + \frac{H_t^{cl}}{\eta_{he}} = 0 \tag{53}$$

In Equations (52) and (53), $P_{gt,t}^{\sigma,RT}$ and $H_{whb,t}^{\sigma,RT}$ are the electrical power of the gas turbine $\sigma$ and the output power of the corresponding waste heat boiler at time $t$ in real-time stage; $H_{b,t}^{\tau,RT}$ is the output power of the gas boiler $\tau$.

*4.3. CVaR Model*

We introduce the conditional value-at-risk (CvaR) of the operation cost as follows:

$$F(x,\alpha) = \alpha + \frac{1}{1-\beta} \sum_{\phi=1}^{N_\phi} \pi_\phi z_k^w \tag{54}$$

$$CVaR = minF(x,\alpha) \tag{55}$$

where CVaR denotes the loss expectation of the detected scenarios in Equation (54) with the auxiliary variable $\alpha$. Auxiliary variable $\alpha$ could be approximated to VaR while CVaR reaches the optimal value [45,46]. In Equation (54), $\beta$ is the confidence level, $N_\phi$ is the number of total scenes, and $\pi_\phi$ is the probability of the occurrence of the $\phi$-th scene; $z_k^w$ is the auxiliary variable at time $t$, which represents the part greater than VaR [47].

The constraints are as follows:

$$z_k^w \geq \mu_1[C_{t,s} + C_{t,waste} + C_{t,w}] + \mu_2 C_{poll,K,t} - \alpha \tag{56}$$

$$z_k^w \geq 0 \tag{57}$$

In Equations (56) and (57), $z_k^w$ is an intermediate parameter with no physical meaning. Then the final objective function considering CvaR is shown as Equation (58):

$$\text{F} = \min\left( f^{DA} + f^{RT} + \lambda CVaR \right) \tag{58}$$

In Equation (58), $\lambda$ represents the relationship between system scheduling cost and risk cost. When $\lambda = 0$, it means that the system operator adopts a risk-neutral strategy,

neither actively avoiding risks nor pursuing risk returns; When $\lambda > 0$, it means that the system operator has begun to gradually avoid operational risks; when the value of $\lambda$ is large, system operators try their best to avoid risks. Therefore, the cost of power generation dispatch is related to the value of $\beta$, and system operators can make reasonable power dispatch decisions based on the degree of risk preference [47].

### 4.4. Solution Strategy

The two-stage optimal scheduling model constructed in the thesis is a mixed integer linear programming problem, which is modeled in MATLAB R2015b using the YALMIP toolbox and called CPLEX for optimal solution. The process is shown in Figure 4.

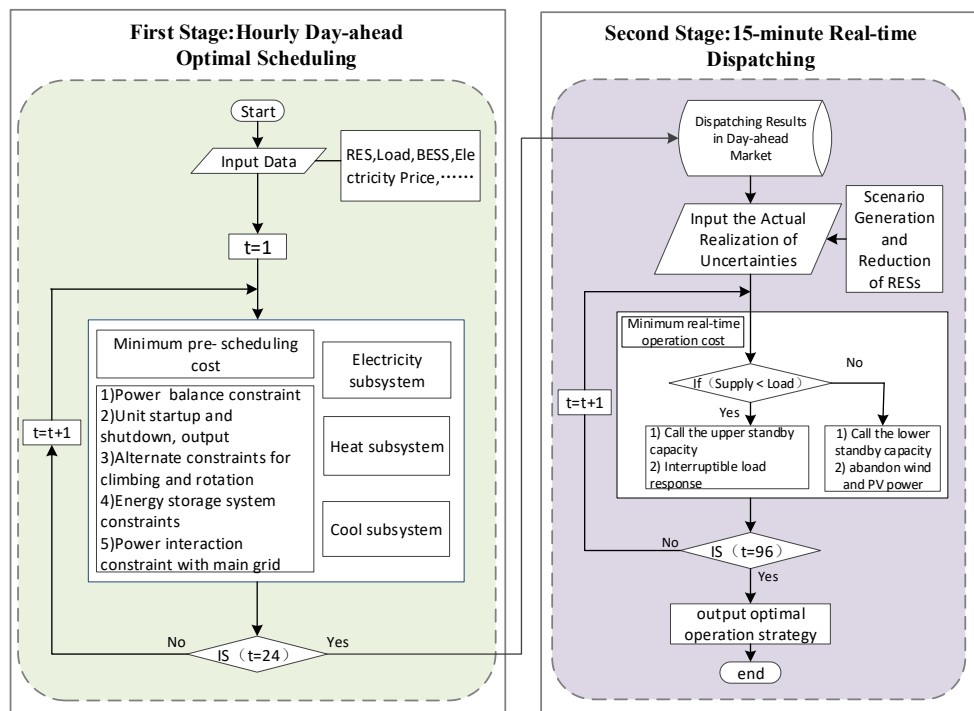

**Figure 4.** Two-stage optimization model solution process.

## 5. Case Studies

### 5.1. Basic Settings

The historical data of wind power and PV power comes from the actual data of a park in Shanxi Province in August 2020, using Monte Carlo simulation to generate 500 sets of PV power scenarios and 500 sets of wind power scenarios. Too many scenarios can complicate the solution, and too few scenarios can affect the accuracy of the results. In order to take into account both the complexity of solution and the accuracy of result, k-means is used to cluster the scene, and 10 typical scenes are obtained. Taking K as parameter, the k-means algorithm divides all objects into K clusters, which makes them have higher similarity in clusters and lower similarity among clusters. The ratio of the number of scenes in the k cluster to the total number of scenes is the probability of the scenes represented by the cluster. The probabilities of typical scenes can be obtained by the above methods. The real-time wind and PV power scenes are shown in Figures 5 and 6 respectively, and the probabilities of ten sets of wind and PV power scenes are shown in Table 1.

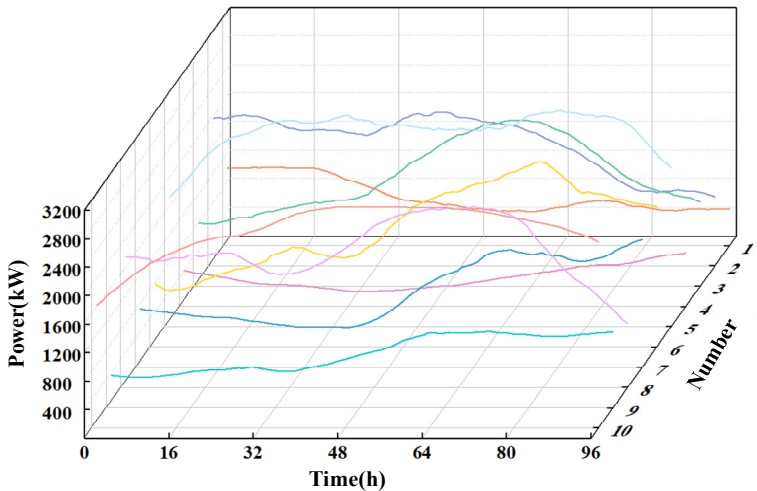

**Figure 5.** Ten sets of wind power scenarios.

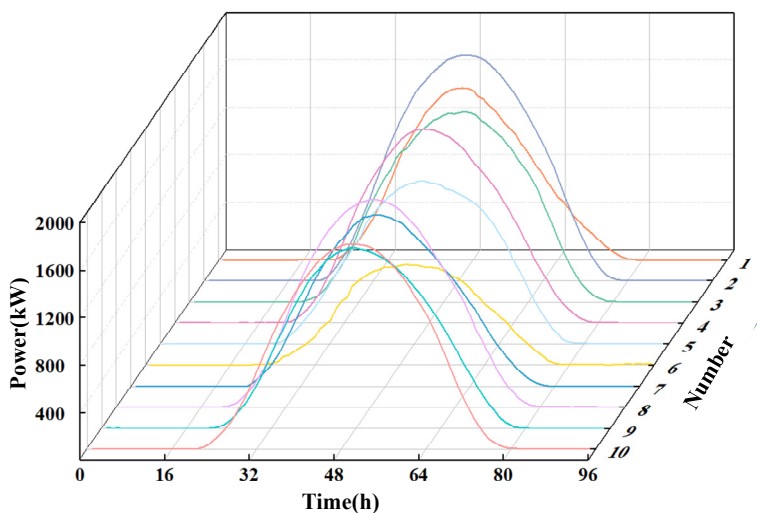

**Figure 6.** Ten sets of PV power scenarios.

**Table 1.** Probabilities of ten sets of wind and PV power scenes.

| PV Power Scenarios | Probability | Wind Power Scenarios | Probability |
|---|---|---|---|
| 1 | 0.058824 | 1 | 0.117647 |
| 2 | 0.117647 | 2 | 0.117647 |
| 3 | 0.117647 | 3 | 0.078431 |
| 4 | 0.098039 | 4 | 0.156863 |
| 5 | 0.098039 | 5 | 0.078431 |
| 6 | 0.058824 | 6 | 0.078431 |
| 7 | 0.098039 | 7 | 0.137255 |
| 8 | 0.156863 | 8 | 0.058824 |
| 9 | 0.117647 | 9 | 0.137255 |
| 10 | 0.078431 | 10 | 0.039216 |

The user load does not consider the specific energy use facilities, and integrates the energy demand of the system into three load curves of electrical, heating and cooling. In order to simplify the calculation process of the model, without considering the fluctuation of the load on the user side, the actual load curve of a park in Shanxi Province in August 2020 is selected for calculation, as shown in Figure 7.

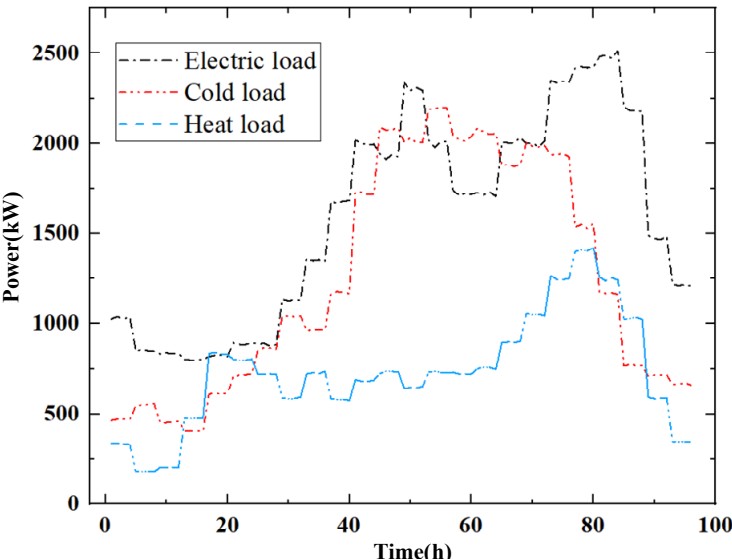

**Figure 7.** Cooling, heating, and electric load curve.

The gas price is 0.53 USD/m$^3$. The hourly electricity price adopts peak-valley electricity price issued by the local government, as shown in Figure 8.

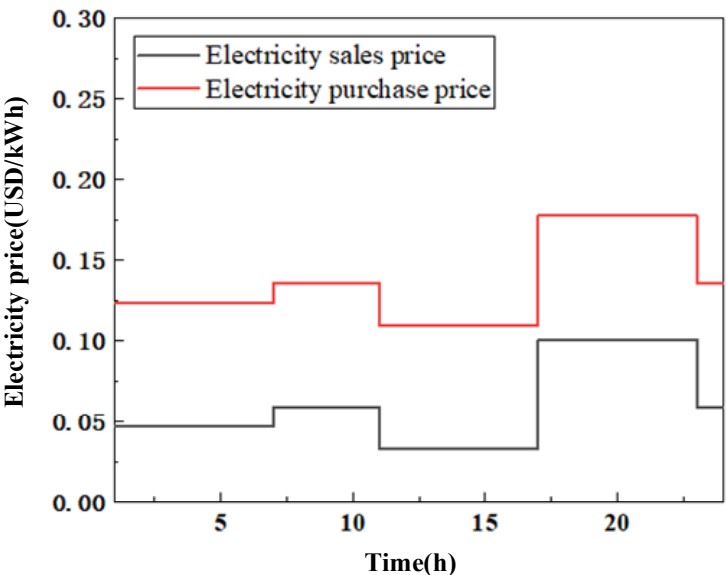

**Figure 8.** Electricity price curve between micro-energy grid and utility grid transaction.

There are three types of gas turbines, with a total of seven types of equipment available for use. There are also three types of gas boilers, one for each. Finally, all other equipment is represented by the last one. See the Tables A1 and A2 for specific information.

*5.2. Results and Discussion*

1. Basic analysis of system energy supply

As shown in Figure 9a, wind power and PV power are given priority as energy sources with zero marginal cost. Between 1:00 and 2:00, the system bought more power from the utility grid because the number of gas turbines started at the initial moment was small and the maximum climbing power was limited. The system must purchase power from the utility grid to ensure the system power supply. From 19:00–21:00, more power was also purchased from the utility grid, which was caused by insufficient power supply in peak load period. During the period from 16:00 and from 23:00–24:00, the electrical load

suddenly drops, the gas turbine is limited by the climbing power and cannot reduce the power quickly. The only way to achieve a balance between supply and demand is through the sale of electricity. In the day-ahead stage, renewable energy, gas turbines, battery storage, and the utility grid jointly meet the power demand of the system. When the real-time power of renewable energy is greater than the day-ahead power, the reserve capacity of the gas turbine and the abandonment of wind and PV power are used to achieve a balance between supply and demand; When the real-time of renewable energy is less than the day-ahead power, the system balance can be satisfied by calling the reserve capacity on the gas turbine and the interruptible load on the user side. As shown in Figure 9b, the system heating load is supplied by the waste heat generated by the gas turbine, the gas boiler, the electric boiler, and the heat storage tank. However, because the cost of electric boilers is higher than that of gas boilers, and the summer heat load is lower, the system does not need to call electric boilers to meet the heat load demand. As shown in Figure 9c, the cooling load of the system is met by the absorption chiller, electric chiller, and ice storage unit. The system preferentially calls the absorption chiller, because the cost of absorption chiller is lower. When the demand cannot be met, the electric chiller is then dispatched for cooling. This conclusion is consistent with the conclusion of the optimal dispatching result of the microgrid proposed by other scholars. They all give priority to the output power of renewable energy. Second, the called units are selected according to the generation cost and climbing constraints of various types of units. When the power still cannot meet the balance, it is necessary to trade with the external power grid [2,9,19–21,38], which verifies the effectiveness of the model in this paper.

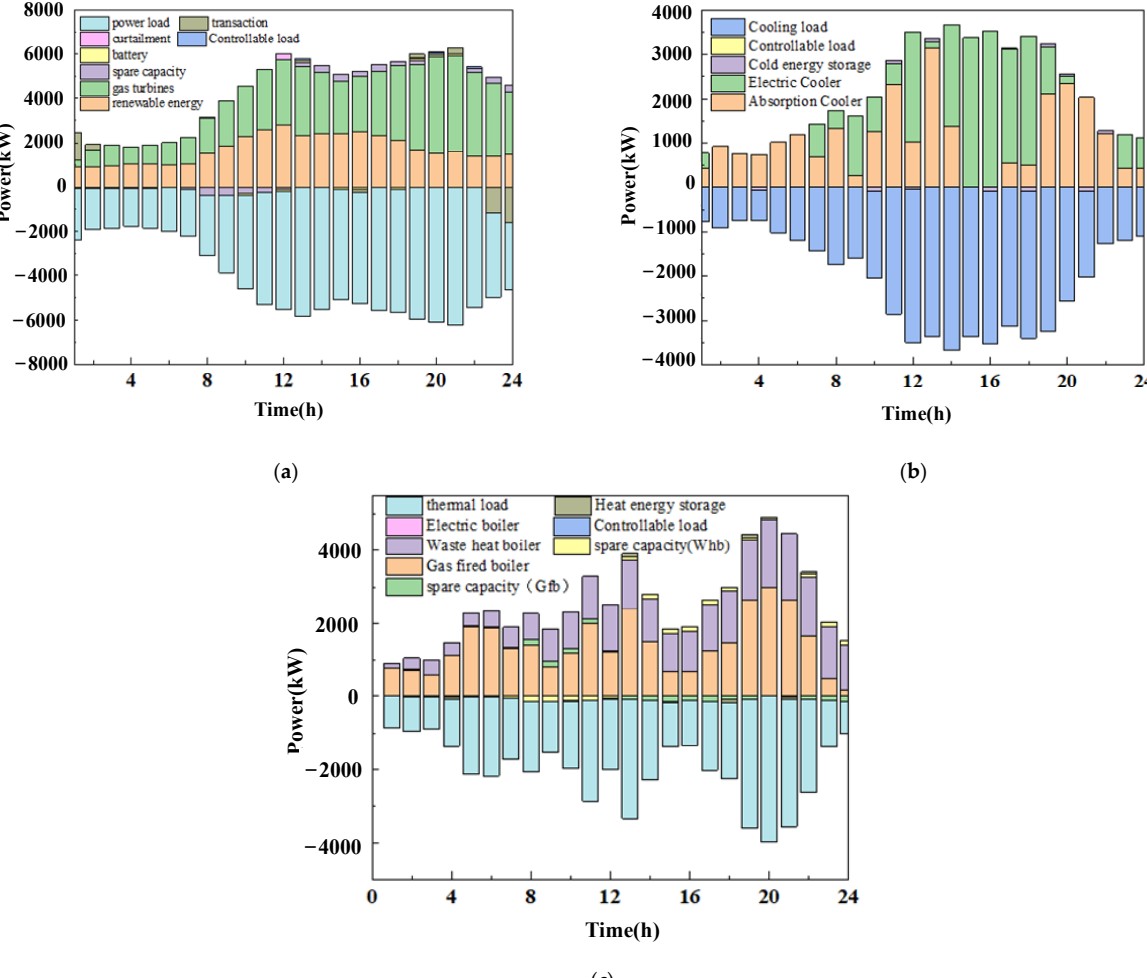

**Figure 9.** (**a**) Electrical power balance; (**b**) heating power balance electrical; (**c**) cooling power balance.

Figure 10a shows that between 0:00 and 12:00, the real-time power of renewable energy is greater than the day-ahead power, and the gas turbine balances the power by calling the down standby capacity. However, in some periods, just scheduling the standby capacity of the gas turbine still cannot absorb the excess RES. Therefore, the phenomenon of wind and PV power abandonment occurs in the three periods of 3:00–5:00, 8:00, and 12:00. For most of the period from 13:00–20:00, the real-time power of wind power is less than the day-ahead forecast; the gas turbine meets the power demand by mobilizing the up standby capacity. At 13:00, 19:00–20:00, and 22:00, the real-time RES is greatly reduced, and at the same time, the gas turbine is insufficiently on standby, which leads to the phenomenon of load shedding. As shown in Figure 10b, the variation of the waste heat of the gas turbine caused by the fluctuation of renewable energy sources in the real-time stage is adjusted by the reserve capacity of gas boiler. The above results indicate that the two-stage optimal scheduling has made sufficient and effective standby arrangements for the uncertainty of renewable energy power.

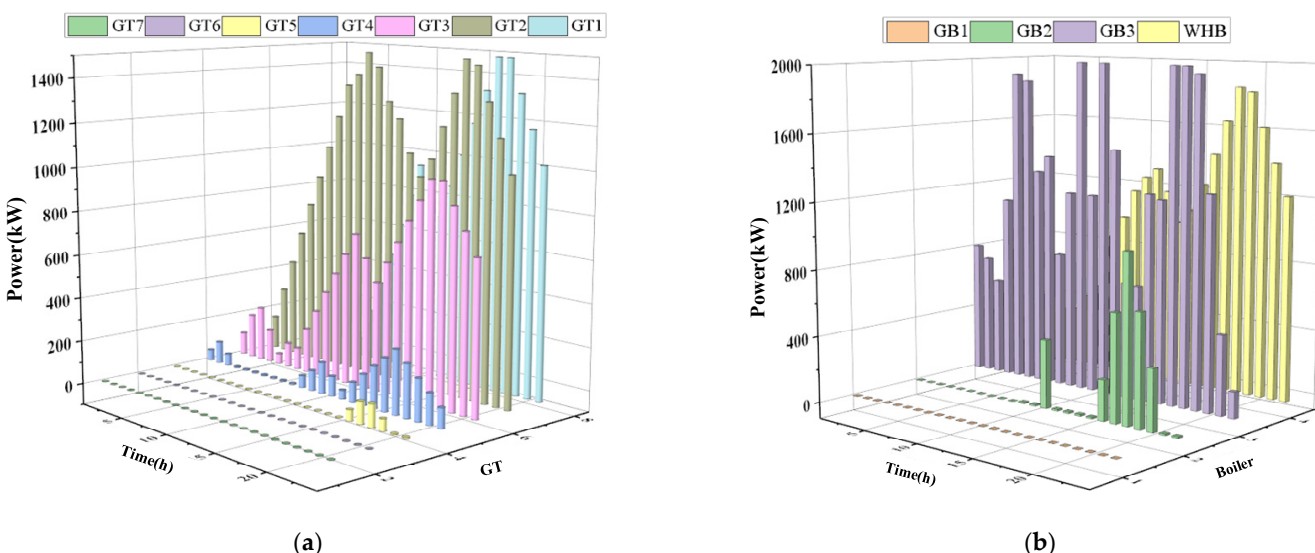

(**a**)　　　　　　　　　　　　(**b**)

**Figure 10.** (**a**) Dispatching power of gas turbines; (**b**) dispatching power of gas boilers.

Figure 11 shows the scheduling of system electrical, heating, and cooling energy storage. The battery is charged from 10:00–12:00 and 15:00–18:00, and discharged at 13:00 and from 19:00–22:00. The battery is to store the electricity when the gas turbine has spare power and discharge it during the peak load period. The results of the heat energy scheduling show that the heat storage tank stores the excess heating energy during the periods of 4:00, 10:00–12:00, 15:00, 18:00, and 21:00, and releases heating energy at 13:00, 19:00–20:00 and 22:00, avoiding energy waste and realizing full utilization of heat energy. The ice storage machine purchases energy to store cooling during the low period of electrical load and heat load, and then releases cooling at 13:00, 17:00, 19:00–20:00, realizing the division of the supply time of the conversion of electricity into cooling energy, thus avoiding the excessively high electricity purchase price of peak load, and saving part of the electricity purchase cost.

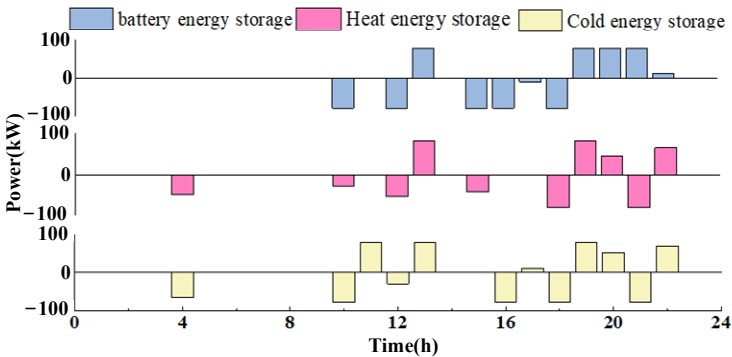

**Figure 11.** Energy storage system dispatching power.

2. The impact of different CVaR values on system scheduling

In order to compare the influence of different risk preferences on the dispatching of heating and electrical units in the system, according to the MILP model, the dispatching of gas turbines and gas boilers under different risk factors is calculated. The confidence level of all test cases is $\alpha = 0.95$. Select three different risk coefficients to compare the impact of risk coefficients on unit scheduling results under different risk preferences: (1) $\lambda = 0$, neither actively avoiding risks nor pursuing risk returns; (2) $\lambda = 2$, adopting strategies to avoid risk, but can still bear a certain risk; (3) $\lambda = 5$, high aversion to risk, and trying to reduce the cost of risk.

Figures 12 and 13 plot the start-up numbers of gas turbines and gas boilers under different risk factors. The shape of the curve and the load curve are similar. During the peak load period, with the increase of the risk factor $\lambda$, the number of gas turbines and gas boilers both show an upward trend. This is because with the increase of $\lambda$, the system gradually avoids the risk of cost increase caused by the fluctuation of renewable energy, so as to use as many units as possible to stabilize the fluctuation of renewable energy. In addition, Figure 12 also shows that during the low load period, the change in the risk coefficient has no impact on unit scheduling. This is because the reserve capacity of gas turbines and gas boilers is sufficient during the low load period, the risk cost caused by the volatility of renewable energy power is little, and the change of the risk coefficient does not change the startup and shutdown of the unit. The difference between Figure 12 is that with the increase of $\lambda$, the number of gas turbine startups also changes; but when $\lambda = 5$, the starting conditions of the gas boiler are the same as when $\lambda = 2$. This is because the calculation example does not consider the fluctuation of the load side, and the risk cost only comes from the fluctuation of the renewable energy power between the day-ahead and the real-time stage, and the fluctuation of the renewable energy has little influence on the heat load supply. Therefore, when $\lambda = 2$, the heat supply unit scheduling plan has minimized the risk cost caused by the interrupted load, and even if the risk factor is further increased, the system heat supply unit scheduling situation will not be changed.

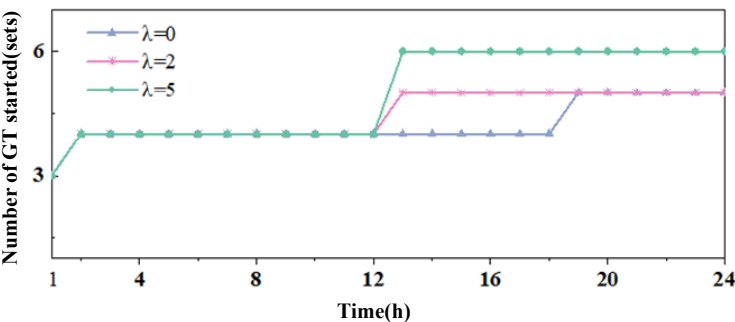

**Figure 12.** Number of start-ups of gas turbines under different risk factors.

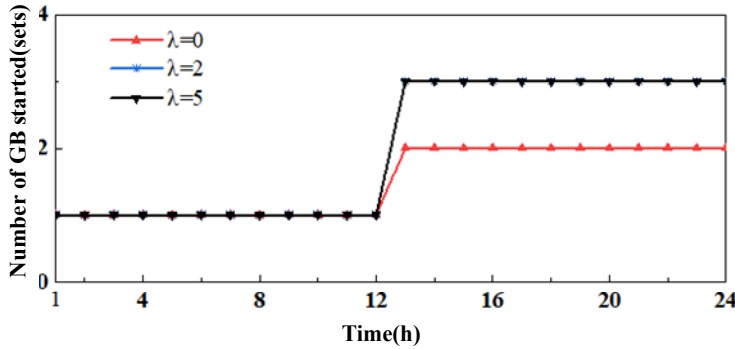

**Figure 13.** Number of start-ups of gas boilers under different risk factors.

Figure 14a,b plot the power of gas turbines and gas boilers under different risk factors. Under the three risk appetites, the power of each unit is different in different time periods. Figure 14a shows that the power of gas turbines 1, 2, and 3 is relatively large, while the output of units 4, 5, 6, and 7 is relatively small. With the increase of risk coefficient, both gas turbines and gas boilers show signs of power transfer from high load rate units to low load rate units. This conclusion is consistent with the conclusion put forward by some scholars that as the risk factor increases, the number of start-ups of thermal power units increases [47], which verifies the effectiveness of the model in this paper.

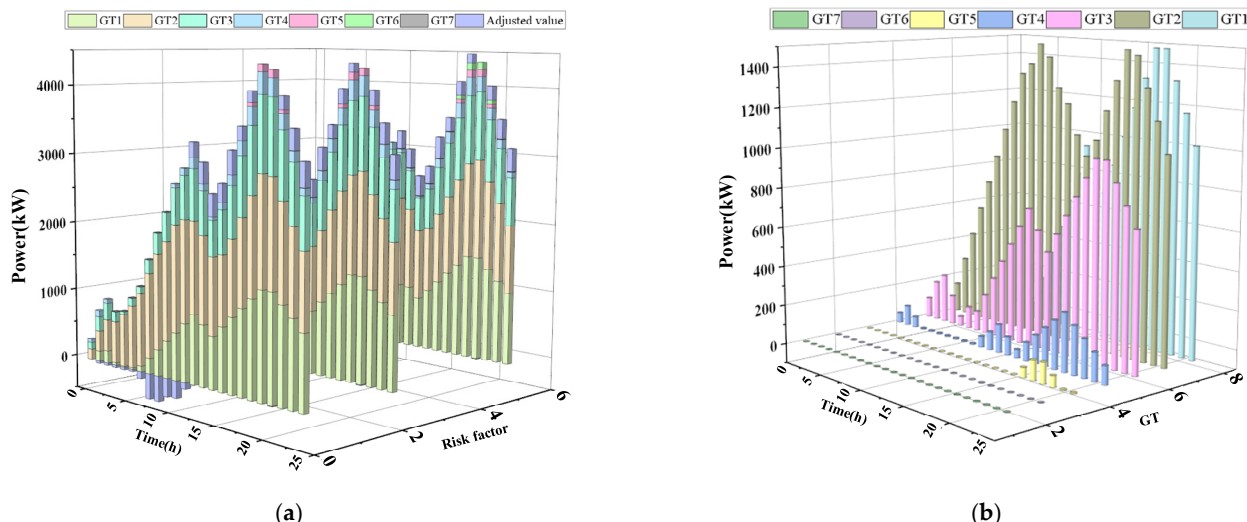

(**a**)          (**b**)

**Figure 14.** (**a**) The power of gas turbines under different risk factors; (**b**) the power of gas boilers under different risk factors.

Table 2 shows the various cost data of the system under different risk factors $\lambda$, including the day-ahead pre-dispatch cost, real-time rescheduling cost, total dispatch cost, and risk cost CVaR. Table 3 shows the amount of wind abandonment, PV abandonment and load shedding of the system under different risk factors $\lambda$. The data in Tables 2 and 3 show that with the increase of $\lambda$, the total dispatch cost of the system increases, but the amount of wind abandonment, PV abandonment and load shedding decrease. That is, when the system dispatcher does not care about the risks caused by the fluctuation of renewable energy sources, a smaller number of gas turbines and gas boilers are started in the day-ahead stage, and a smaller reserve capacity is used to meet the supply and demand balance of real-time operation; this also leads to the amount of wind abandonment, PV abandonment, and interruptible load being relatively high in the real-time stage. When the system dispatcher tries to reduce the risk cost, the performance is to increase the number of units to start in the day-ahead phase, but there will be a situation where the units are in a low load rate operation state for the reserve capacity; and the phenomenon of load shedding, wind abandonment and PV abandonment during real-time operation

will be reduced. In addition, it can be seen from Table 2 that with the increase of $\lambda$, the number of system units dispatched increases, leading to an upward trend in total dispatch costs, while CVaR shows a downward trend due to the reduction of wind abandonment, PV abandonment, and load shedding. Therefore, it is impossible to pursue the further improvement of another goal without damaging one goal between the expected total cost and CVaR. In the actual decision-making, the specific value of risk coefficient should also be determined in combination with the subjective preference of the system dispatcher. This conclusion is consistent with the previous conclusion that as the risk coefficient increases, the system's scheduling cost also increases [28,30,32,47], which verifies the effectiveness of the model in this paper.

**Table 2.** System scheduling cost under different risk factors.

| $\lambda$ | Day-Ahead Pre-Dispatching Cost/\$ | Real-Time Rescheduling Cost/\$ | Total Dispatch Cost/\$ | CVaR/\$ |
|---|---|---|---|---|
| 0 | 14,904 | 831 | 15,736 | 19,137 |
| 2 | 14,972 | 810 | 15,782 | 19,081 |
| 5 | 15,025 | 844 | 15,870 | 18,985 |

**Table 3.** Abandoned wind and light and load reduction under different risk factors.

| $\lambda$ | Amount of Abandoned PV/kWh | Amount of Abandoned Wind/kWh | Load Shedding/kWh |
|---|---|---|---|
| 0 | 9.74 | 62.86 | 43.26 |
| 2 | 4.76 | 31.48 | 0 |
| 5 | 0 | 0 | 0 |

3. Environmental analysis

This section discusses the relationship between scheduling results that aim at minimizing the operating cost of the MEG and scheduling results that aim at minimizing the cost of carbon emissions. $\mu_1$ and $\mu_2$, mentioned above, are corresponding weights of operation cost and environment cost respectively. In the actual operation of the MEG, the scheduling cost and environmental cost are unavoidable, that is, the MEG cannot be run without considering any cost. Therefore $\mu_1$, $\mu_2 \in (0,1)$, and $\mu_1 + \mu_2 = 1$. Three scenarios are set up by adjusting the weighting coefficients of system operation cost and environment cost. The relationship between operation cost and system carbon emission is discussed, and the effectiveness of the proposed model is verified. In order to eliminate the influence of risk preference coefficient on system operation, the risk coefficients of the three scenarios are all zero, that is, neither actively avoid risk nor pursue risk return. The scenario settings are shown in the Table 4.

**Table 4.** Parameter settings of three cases.

| Case | $\mu_1$ | $\mu_2$ | $\lambda$ |
|---|---|---|---|
| Case1 | 0.9 | 0.1 | 0 |
| Case2 | 0.1 | 0.9 | 0 |
| Case3 | 0.5 | 0.5 | 0 |

The scheduling principle of scenario 1 is cost-driven, and the economic cost weight of the system is higher than the carbon emission weight. In this scenario, the system dispatcher pursues the minimum dispatching cost, including the dispatching cost of units, gas cost, and the cost of purchasing electricity from the utility grid.

The scheduling principle of scenario 2 is environmentally driven, and the economic cost weight of the system is lower than the carbon emission weight. In this scenario, system dispatchers pursue minimum system carbon emissions, including $CO_2$ from the operation

of gas turbines, gas boilers, and carbon emissions from electricity purchased from the utility grid.

The scheduling principle of scenario 3 is cost-environmental equilibrium scheduling, and the system economic cost weight is consistent with the carbon emission weight. In this scenario, the system dispatcher will balance the system dispatching costs against the $CO_2$ emissions.

As shown in Figure 15, the system in Scenario 1 has the lowest overall economic cost and the highest carbon emissions. Scenario 2 has the lowest emissions, but the highest overall economic cost. The scheduling result of Scenario 3 is more balanced, and the carbon emission and system scheduling cost are in the middle level. It has been proposed that economic goals and environmental goals are two conflicting goals, and neither goal can be improved without lowering other goals [48]. In the process of micro-grid scheduling, the reduction of carbon emissions will inevitably lead to the increase of overall scheduling costs, but the system dispatcher can choose an equilibrium solution. The results show that high carbon emissions-weighted scheduling can reduce carbon trading costs compared with equal weights scheduling, but will increase the operating costs of high carbon emissions-weighted scheduling [49]. The above conclusions are consistent with the results of this paper, and verify the effectiveness of the proposed model considering economic and environmental protection. The results show that the weighting coefficient of system operation cost and environmental cost has a significant impact on system economy and carbon emission in MEG scheduling. Therefore, in the actual scheduling process, system dispatchers can determine different environmental cost weight coefficients according to carbon emission preference.

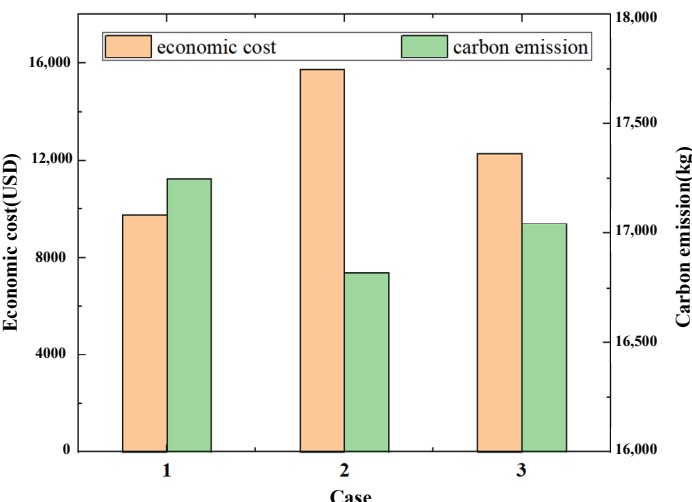

**Figure 15.** Cost and emission comparison for three cases.

## 6. Conclusions

This paper presents a two-stage MEG planning model considering CVaR aggregated with a high proportion of RES and multiple types of energy consumption. The proposed model effectively co-optimizes scheduling strategy of multi-energy consumption of MEG, as well as the assessment on CVaR. The objective function of this model covers the day-ahead and real-time scheduling cost, pollutant emission cost, and risk cost caused by the uncertainty of RES output of MEG. Considering the above objectives comprehensively, the scheduling strategy of micro-energy network is optimized. The planning decisions, including operation strategies for multi-energy device, charging/discharging ESS, and scheduling plan of interruptible load demand response, are optimized in the model. In terms of risk assessment, a CVaR based analysis is implemented in different risk preference.

The case studies demonstrate the effectiveness of the proposed model when applied to multi-energy systems, and illustrate the benefits of CVaR in dealing with the uncertainty

of RES. ESSs located on the demand side can benefit system operation through peak-valley load shifting and energy arbitrage to enhance resilience. In addition, the system scheduling cost can be minimized, and the risk of energy shortage caused by the randomness of renewable energy can be avoided to the greatest extent, by adjusting the risk preference coefficient. In this way, the goal of the highest user energy quality and the minimum system scheduling cost can be achieved.

**Author Contributions:** Conceptualization, J.D. and Y.Z.; methodology, Y.Z.; software, Y.Z. and Y.W.; validation, J.D., Y.Z. and Y.W.; formal analysis, Y.Z.; investigation, Y.L.; resources, Y.L.; data curation, Y.L.; writing—original draft preparation, Y.Z.; writing—review and editing, Y.Z. and Y.W.; visualization, Y.W.; supervision, J.D.; project administration, J.D. All authors have read and agreed to the published version of the manuscript.

**Funding:** This research received no external funding.

**Institutional Review Board Statement:** Not applicable.

**Informed Consent Statement:** Not applicable.

**Data Availability Statement:** Not applicable.

**Conflicts of Interest:** The authors declare no conflict of interest.

## Abbreviations

| | |
|---|---|
| MEG | Micro Energy Grid |
| RES | Renewable energy sources |
| VaR | value-at-risk |
| CVaR | Conditional value-at-risk |
| MILP | mixed integer linear programming |
| ESS | energy storage system |
| CL | controllable load |
| PV | photovoltaic |
| WT | wind turbine |
| GT | Gas turbine |
| GB | Gas boiler |
| WHB | waste heat boiler |
| EB | electric boiler |
| EC | electric chiller |
| AC | absorption chiller |

## Appendix A

**Table A1.** Gas turbine equipment parameters.

| Type | Rated Capacity | Electrical Efficiency | Thermal Efficiency | Start Stop Cost | Operation and Maintenance Cost | Upper and Lower Limits of Power | Climbing Power | Initial State |
|---|---|---|---|---|---|---|---|---|
| 1 | 1500 | 0.28 | 0.54 | 69 | 0.01 | 1500/15 | 620 | 1 |
| 2 | 1500 | 0.31 | 0.50 | 68 | 0.01 | 1500/15 | 620 | 1 |
| 3 | 1000 | 0.24 | 0.52 | 46 | 0.01 | 1000/10 | 410 | 1 |
| 4 | 500 | 0.2 | 0.45 | 23 | 0.01 | 500/5 | 220 | 1 |
| 5 | 500 | 0.2 | 0.45 | 23 | 0.01 | 500/5 | 220 | 0 |
| 6 | 500 | 0.2 | 0.45 | 23 | 0.01 | 500/5 | 220 | 0 |
| 7 | 500 | 0.2 | 0.45 | 23 | 0.01 | 500/5 | 220 | 0 |

**Table A2.** Gas boiler equipment parameters.

| Type | Rated Capacity | Efficiency | Start Stop Cost | Operation and Maintenance Cost | Upper and Lower Limits of Power | Climbing Power | Initial State |
|---|---|---|---|---|---|---|---|
| 1 | 600 | 0.63 | 25 | 0.045 | 600/6 | 200 | 0 |
| 2 | 800 | 0.75 | 32 | 0.045 | 800/8 | 300 | 1 |
| 3 | 1000 | 0.88 | 40 | 0.045 | 1000/10 | 450 | 1 |

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
