# Peer review of "A Two-Stage Optimal Dispatching Model for Micro Energy Grid Considering the Dual Goals of Economy and Environmental Protection under CVaR"

_sustainability, doi:10.3390/su131810173_

Round 1

Reviewer 1 Report

In this paper, Jun et al describes the use of two-stage optimal scheduling model that takes into account both economic and environmental impacts to solve the problem of micro energy grid. They measure the risk using CvAR (conditional value at risk).

Overall this paper has some value to the community, although the model is not exactly very new.

The authors must address the following before it can be recommended for publication.

  1. There seems to be no time-dependency in the power equation of PV (equation 2), at night, there should be no power. Has this been taken into account in the model?
  2. in equation 12, the heating energy storage, the authors seems to neglect heat loss due to imperfect insulation. This can be crucial considering no amount of insulation can really prevent heat loss. How do the authors reconcile this?
  3. How did the authors arrive at the probability number in table 1 is not clear at all. Please elaborate.
  4. the time grid in figure 7 is to rough. (1 grid seems to represent 1h), how will this affect the accuracy?
  5. On a related note, it is meaningless to claim number such as "the optimal operating cost of the system is 34873.47 USD. Please round it to the nearest 10000 figure.
  6. In figure 14, the risk factor axis is not clear at all.
  7. figure 15 is not well explained, what are the boundary conditions?

The authors should carefully address the issues before it can be recommended for publication.

Author Response

Point 1: There seems to be no time-dependency in the power equation of PV (equation 2), at night, there should be no power. Has this been taken into account in the model?

Response 1: Thank you for your comment.  is used to describe PV power. It can be seen that photovoltaic power is determined by the product of the actual solar irradiance (ζ), the solar area of the photovoltaic panel ( ) and the photoelectric efficiency of the photovoltaic panel ( ). At night, the solar irradiance is 0, so the power of photovoltaic is 0. please see the revisions using “Track Changes” functions in Section 3.1.

Point 2: In equation 12, the heating energy storage, the authors seems to neglect heat loss due to imperfect insulation. This can be crucial considering no amount of insulation can really prevent heat loss. How do the authors reconcile this?

Response 2: Thank you for your comment.  is used to describe the heat loss due to imperfect insulation of the heat storage tank. The description of the heat storage tank model is not clear. We have revised the description of the heat storage tank model and reinterpreted it as follows.

In Equation (12),  and  respectively represent the capacity of the heat energy storage device at t and t-1.  and  respectively represent the heating storage and release power of the heating energy storage device. ,  and  represent heat loss rate and charge/discharge efficiency of heating energy storage, respectively. please see the revisions using “Track Changes” functions in Section 3.3.

Point 3: How did the authors arrive at the probability number in table 1 is not clear at all. Please elaborate.

Response 3: Thank you for your comment. We have specified the sources of the probabilities in Table 1 as follows.

The historical data of wind power and PV power comes from the actual data of a park in Shanxi Province in August 2020, using Monte Carlo simulation to generate 500 sets of PV power scenarios and 500 sets of wind power scenarios. Too many scenarios can complicate the solution, and too few scenarios can affect the accuracy of the results. In order to take into account both the complexity of solution and the accuracy of result, k-means is used to cluster the scene, and 10 typical scenes are obtained. Taking K as parameter, the k-means algorithm divides all objects into K clusters, which makes them have higher similarity in clusters and lower similarity among clusters. The ratio of the number of scenes in the k cluster to the total number of scenes is the probability of the scenes represented by the cluster. The probabilities of typical scenes can be obtained by the above methods. The real-time wind and PV power scenes are shown in the figure 5 and figure 6 respectively, and the probabilities of ten sets of wind and PV power scenes are shown in Table 1, please see the revisions using “Track Changes” functions in section 5.1.

Point 4: The time grid in figure 7 is to rough. (1 grid seems to represent 1h), how will this affect the accuracy?

Response 4: Thank you for your comment. We have modified Figure 7 to a load curve of 96 points to make the calculation more accurate, please see the revisions using “Track Changes” functions in Figure 7.

Point 5: On a related note, it is meaningless to claim number such as "the optimal operating cost of the system is 34873.47 USD. Please round it to the nearest 10000 figure.

Response 5: Thank you for your comment. We have round it to the nearest 10000 figure, please see the revisions using “Track Changes” functions.

Point 6: In figure 14, the risk factor axis is not clear at all.

Response 6: Thank you for your comment. We have modified the risk factor axis in Figure 14 to make the results look clearer, please see the revisions using “Track Changes” functions in Figure 14.

Point 7: Figure 15 is not well explained, what are the boundary conditions?

Response 7: Thank you for your comment.  and  mentioned above are corresponding weights of operation cost and environment cost respectively. In the actual operation of the MEG, the scheduling cost and environmental cost are unavoidable, that is, the MEG can not be run without considering any cost. Therefore  and .

Figure 15 in the paper and its corresponding explanation is not perfect, we have made a more detailed changes to this section. In order to explore the relationship between system scheduling cost and carbon emissions, the cost-oriented scenario ( ), environment-oriented scenario ( ) and balanced scenario ( ) were re-established. The change trend of operating cost and carbon emission under different operating cost weights and environmental cost weights was analyzed. The conclusion is drawn that the decrease of carbon emission will lead to the increase of overall dispatching cost of the system. This conclusion is consistent with the conclusions in other references, and the validity of the proposed model is verified, please see the revisions using “Track Changes” functions in section 5.2.

Ends.

Reviewer 2 Report

The authors proposed a two-stage model with optimization on the economy and environmental protection. The authors could have more discussion in the Section 5 by comparing and validating their results with references. Compared to others' work, what are the advantages and challenges? For example, in the third part of Section 5, is the CO2 emissions relation validated by the industry or published data? Is there related work that shows similar trend? More references are required. 

Author Response

Point: The authors proposed a two-stage model with optimization on the economy and environmental protection. The authors could have more discussion in the Section 5 by comparing and validating their results with references. Compared to others' work, what are the advantages and challenges? For example, in the third part of Section 5, is the CO2 emissions relation validated by the industry or published data? Is there related work that shows similar trend? More references are required.

Response: Thank you for your comment. The paper is revised as follows:

  • In terms of system scheduling results, the best scheduling results of microgrids in previous studies are all prioritizing renewable energy output power. Secondly, the units to be called are selected based on the power generation cost and the climbing constraints of each type of unit. When the power still cannot meet the balance, the system needs to be traded with the external grid or choose to abandon wind and PV power to make the system achieve power balance, which is consistent with the results of this paper. For specific analysis, see the first part of Subsection 5.2.
  • In terms of system operation risk, some scholars have proposed that as the risk factor increases, the scheduling cost of the system will also increase, and the start and stop status of the unit will also change. This conclusion is consistent with the optimization results of this paper. For specific analysis, see the second part of Subsection 5.2.
  • In terms of the relationship between the operating cost of the micro energy grid system and carbon emissions, three scenarios were set up to analyze the changing trends of operating costs and carbon emissions under different operating costs and environmental cost weights. At the same time, other scholars' research on the relationship between microgrid scheduling costs and carbon emissions is analyzed, and their conclusions are consistent with the conclusions of this paper. For specific analysis, see the third part of Subsection 5.2.

Round 2

Reviewer 1 Report

The authors have sufficiently addressed all my comment. The manuscript can now be recommended for publication.